# Harnessing Machine Learning in Vocal Arts Medicine: A Random Forest Application for “Fach” Classification in Opera

**DOI:** 10.3390/diagnostics13182870

**Published:** 2023-09-06

**Authors:** Zehui Wang, Matthias Müller, Felix Caffier, Philipp P. Caffier

**Affiliations:** 1Institute for Digital Transformation, University of Applied Sciences Ravensburg-Weingarten, Doggenriedstraße, 88250 Weingarten, Germany; zehui.wang@rwu.de; 2Occupational College of Music BFSM Krumbach, Mindelheimer Str. 47, 86381 Krumbach, Germany; matthias.mueller@stimmkultur.de; 3School of Computing, Communication and Business, HTW Berlin University of Applied Sciences, Treskowallee 8, 10318 Berlin, Germany; broozar@web.de; 4Department of Audiology and Phoniatrics, Charité—Universitätsmedizin Berlin, Campus Charité Mitte, Charitéplatz 1, 10117 Berlin, Germany

**Keywords:** vocal arts medicine, voice classification, machine learning, random forest, dramatic voice structure, lyric voice structure, voice timbre parameter, opera singer, digital sound analysis, voice disorder prevention

## Abstract

Vocal arts medicine provides care and prevention strategies for professional voice disorders in performing artists. The issue of correct “Fach” determination depending on the presence of a lyric or dramatic voice structure is of crucial importance for opera singers, as chronic overuse often leads to vocal fold damage. To avoid phonomicrosurgery or prevent a premature career end, our aim is to offer singers an improved, objective fach counseling using digital sound analyses and machine learning procedures. For this purpose, a large database of 2004 sound samples from professional opera singers was compiled. Building on this dataset, we employed a classic ensemble learning method, namely the Random Forest algorithm, to construct an efficient fach classifier. This model was trained to learn from features embedded within the sound samples, subsequently enabling voice classification as either lyric or dramatic. As a result, the developed system can decide with an accuracy of about 80% in most examined voice types whether a sound sample has a lyric or dramatic character. To advance diagnostic tools and health in vocal arts medicine and singing voice pedagogy, further machine learning methods will be applied to find the best and most efficient classification method based on artificial intelligence approaches.

## 1. Introduction

In addition to the artistic aspect, professional singing can also be considered a high-performance sport for the larynx, and just like in high-performance athletes, professional voice care is extremely important [1,2]. Phoniatrics or “voice medicine” is a highly specialized medical field that deals with impairments in the voice in terms of diagnosis, therapy and science [3,4]. Since human vocal function is very complex, the protocol of the European Laryngological Society requires the use of various objective and subjective parameters to describe a voice as accurately and comprehensively as possible [5]. The established diagnostic examination tools are (1) auditory-perceptual voice assessment [6,7,8,9], (2) videolaryngostroboscopy [10,11,12,13], (3) voice range profile measurements for the assessment of vocal performance [14,15,16,17], (4) acoustic-aerodynamic analyses for the evaluation of vocal impairment [18,19,20,21], as well as (5) self-assessment of the voice [22,23,24,25]. Vocal arts medicine is using these diagnostic tools to care for and prevent professional voice disorders in performing artists [26]. In cases of recurrent or persistent voice problems, phoniatric specialists are often confronted with unsolved classification questions: What is the right voice type and “Fach”, lyric or dramatic [27,28]? Since the above-mentioned established examination tools do not include a connection to fach-specific requirements, the precise causes of diagnosed voice problems are currently often difficult to establish adequately.

To date, there are no internationally published studies (apart from [27,28]) within the fields of voice science, pedagogy, or engineering on the objective distinction between lyric and dramatic fach. The scientific literature search revealed only one paper using a ML approach to investigate a timing-based classification method for human voice in opera recordings [29]. However, the aim was merely an automatic identification of famous tenors. In the reality of singing voice pedagogy, the fach question is traditionally answered with an overall view on the respective artist’s personality based on experience and intuition. This traditional approach includes not only the purely vocal requirements, but also all other aspects needed for the realization of an opera role in the corresponding fach. Of course, such a subjective, intuitive approach is also prone to error. However, the issue of correct fach determination is of crucial importance for opera singers to prevent dysodia and vocal fold damage [30,31,32,33]. The German fach system described in detail by Kloiber has occasionally been questioned for some years [34]. In 2008, Ling called it anachronistic and presented a new fach classification for male voices, using “voice structure” [35] (pp. 27–29, 66) and “voice category” [35] (pp. 257–258), among others, as a source of information. He wrote in his proposed solution, “The distinction between lyric and dramatic Fächer … could be retained …” [35] (p. 258) because it primarily describes voice structures to which lyric and dramatic roles can be easily assigned. The present study classifies precisely according to this voice structure, which can be regarded as a necessary prerequisite for mastering a particular fach. In 2016, the editors of the latest edition of Kloiber’s renowned opera guide also decided to reintroduce the critically discussed “Kloiber system” in the detailed chapters on “Fachpartien” *[fach-specific roles]* and “Besetzungsfragen” *[casting concerns]*, restoring central attention to the distinction between lyric and dramatic Fächer [34] (p. 925). Thus, in addition to the artistically motivated discussions of historical performance practice and contemporary stage direction, the purely physical strain required by a lyric or a dramatic role is given the necessary consideration. In a dramatic role, high volume levels with great carrying power and projection are often required for long periods of time. In a lyric role, the average volume of the accompanying orchestra is lower; it is rather a matter of a special sound aesthetic, often a high agility of the voice.

Misclassification and chronic overuse leading to diagnostically confirmed voice disorders are associated with functional, physical and psycho-emotional impairment, often resulting in a shortened career, occupational disability or phonomicrosurgery [36,37,38,39,40]. Top-level vocalists are therefore not only coached by singing teachers [41], but also supported by phoniatricians or otorhinolaryngologists with voice specialization [2,42]. This interdisciplinary cooperation is not only useful for preventive, diagnostic and therapeutic reasons in individual singers, but is also suitable to answer scientifically still unclear, singing-associated questions. However, this requires examining large amounts of data from a high number of singers, making the inclusion of machine learning (ML) methods extremely helpful. The central and innovative goal of the present work is to offer singers an improved, objective fach, counseling using digital sound analyses and ML procedures, to advance diagnostic tools and health in vocal arts medicine and singing voice pedagogy.

## 2. Materials and Methods

This section describes our interprofessional methodological approach, starting from the initial musical score studies for the identification of suitable phrases via a selection of sound samples from complete recordings of the respective operas, the construction of the database, definition of appropriate parameters for classification, and up to the final steps of statistical analyses and ML application. To introduce and better understand the process, the workflow chart in Figure 1 gives a general overview of the applied procedures, which will be explained in more detail below.

The extensive database contains sound samples of healthy, well-known opera singers performing in either the lyric or dramatic fach. For the digital fach classification, 281 mezzo-soprano sound samples were added to the 1723 originally collected data. As of 30 June 2023, our database contained a total of 2004 sound samples for the voice types soprano (*n* = 774), mezzo-soprano (*n* = 281), tenor (*n* = 389), baritone (*n* = 343) and bass (*n* = 217). In terms of voice structure, our cohort was classified as follows: 422 dramatic vs. 352 lyric sopranos, 147 dramatic vs. 134 lyric mezzo-sopranos, 190 dramatic vs. 199 lyric tenors, 186 dramatic vs. 157 lyric baritones, and 157 dramatic vs. 60 lyric basses.

The sounds have a duration of 6 up to 15 s and are encoded using 16-bit PCM samples at a rate of 22,050 Hz. The criteria for selecting the sounds included the following: (1) a single note in musical notation, (2) no vocal changes and (3) no orchestral accompaniment. Acoustic analysis was limited to this one tone; however, each sample contains a few seconds of musical context before and after the analysis interval. The sound samples were extracted from commercially available complete recordings of the operas. Within each voice type, the sound samples were labeled either as dramatic (1) or as lyric (2) based on the musical role from which the samples were extracted. Kloiber [34] and Ling [35] (pp. 257–286) were used as established references for this attribution. In addition to the lyric or dramatic voice structure, the samples were labeled with further codes containing the following information: voice type, gender, name of singer, musical part, vowel and musical pitch. Labeling examples for selected sound samples in every investigated voice type are shown in Figure 2. A more detailed description of the database can be found in a previous study [27].

For the analysis of the sound files, a development environment was programmed with the software MATLAB version R2017b (The MathWorks, Inc., Natick, MA, USA) which can automatically calculate various features of the sound files. The expanded database makes it possible, for the first time, to present meaningful analysis data of professional opera voices on this scale. Based on the labels assigned in the file names, the system learns which objectively recognizable characteristics a lyric or a dramatic sound sample has.

In order to facilitate the application of ML techniques for resolving the fach classification issue, a collection of 13 distinctive features were meticulously computed and extracted from the original sound files. These features include the following:Start and Stop of Formant (StaFo and StoFo), providing insight into the frequency range that most significantly amplifies vocal tract resonance;Position of Half Energy (PHE) and Frequency of Half Energy (FHE), offering valuable information about the spectral balance of the voice by representing the midpoint position of the energy distribution;Vibrato Rate (VR) and Vibrato Extent (VE), which elucidate details about the speed and extent of pitch oscillation;Perturbation measures Jitter (Jitt) and Shimmer (Shim), which quantify voice stability by calculating cycle-to-cycle variations of the fundamental frequency and amplitude;Strength of Formant Energy (StreFo), signifying the energy intensity within the singer’s formant frequency bands;Further spectral features, i.e., Spectral Centroid (SC), Variance (SV), Skewness (SS), and Kurtosis (SK), delivering essential data about the distribution of energy in the singer’s formant frequency bands.

For a detailed definition of these features, please refer to our previous publications [27,28].

Descriptive statistics were used to determine the basic characteristics of the data. For each voice type and fach, minimum and maximum values, medians, quartiles, means (M), standard deviations (SD) and reference ranges (RR = M ± 1.96 × SD) were computed. To keep the statistical tests consistent throughout all categories, non-parametric tests (which do not rely on normality) were used whenever possible. The Mann–Whitney U test for independent samples was applied because a normal distribution of the data could not be established in each voice type. Histograms were plotted to display the timbre data for fach-specific differences. Pearson correlation coefficients for interval-scaled variables were calculated and visualized in a heat map to identify the relationships between features and to test for collinearity. All statistical analyses were performed using SPSS version 28.0.1 (IBM, Armonk, NY, USA). The level of significance was set at α = 0.05.

The numerical features were fed into a ML classifier in order to distinguish between lyric and dramatic categories based on these specific voice characteristics. The Random Forest (RF) procedure was used for fach classification, an ensemble learning algorithm conceived by Breiman in 2001 [43]. The underlying concept of RF is based on the wisdom of crowds. It includes numerous decision trees and implements random feature and data selection during the training phase. The results from each decision tree are combined, leading to the final decision. Because the RF algorithm is considered one of the most proficient algorithms in the domain of ensemble learning [44], it was the optimal choice for our task. To classify vocal characteristics into lyric or dramatic voice structures, a RF classifier from the Scikit-Learn library (scikit-learn version 1.2.2) [45] was harnessed within a Python environment (Python version 3.9, Python Software Foundation, Wilmington, DE, USA). For the purpose of enhancing the robustness and generalizability of our ML model, we leveraged a stratified 10-fold cross-validation process. This approach ensured an equitable representation of each class across the folds, thereby mitigating potential biases in the training set. This careful methodology should provide reliable classification results in order to objectively delineate between lyric and dramatic voices. The pseudocode-like algorithm used to obtain the classification results is presented in Table 1. The corresponding Python source code is linked in the Appendix A.

Subsequently, the ML model calculations were scrutinized based on the confusion matrix shown in Table 2.

The efficacy of the model was ascertained using the metrics of accuracy and Balanced Error Rate (BER) [46]. Accuracy describes the performance of a classification model as the number of correct predictions (both true positives and true negatives) divided by the total number of cases examined:(1)$$Accuracy=\frac{TP+TN}{TP+TN+FP+FN}$$

With perfect accuracy, the formula equals 1 (i.e., 100% of correct classifications). However, achieving 100% ML model accuracy is typically a sign of some error, such as overfitting. On the other hand, the BER balances both possible error rates by calculating the average between the *FP* rate and *FN* rate:(2)$$BER=0.5\times \left(\frac{FP}{TN+FP}+\frac{FN}{TP+FN}\right)$$

The further the result tends towards 0 (or 0%), the better the *BER*.

Finally, to explore the role of features in the RF classifier, the importance of features during the classification process was determined for each voice type. To provide an explanation of the predictions of the RF classifier, we used SHapley Additive exPlanations (SHAP). This tool for visualizing ML models measures how much each feature affects the prediction, i.e., with SHAP we can understand whether an increase in the value of the feature improves or worsens the prediction.

## 3. Results

As expected, descriptive statistics provided summarizing information of all vocal parameters and shed light on the distribution of values in our datasets. Table 3 shows the means, SD and 95% reference rages (RR) for the recently introduced timbre parameters in all voice types and for both voice structures (lyric vs. dramatic).

In general, the mean values for PHE, FHE and SC were larger for the higher voice types than for the lower ones. However, the 95% RR revealed a substantial overlap between adjacent voice types within females and males. A difference between both voice structures was determined within each voice type: lyric sopranos, tenors and baritones had significantly higher timbre parameters than dramatic ones (*p* < 0.001). For the mezzo-soprano and bass voices, the mean differences regarding voice structure were much smaller and statistically not significant. These relations between the abstract *p*-values and the timbre parameters were also audible: the larger the perceptible discrepancy of the vowel-independent timbre between both voice structures, colloquially referred to as brightness, the more likely this difference is statistically significant.

The histograms of PHE and SC sorted by voice type and fach are shown in Figure 3.

The correlations among all 13 features were successfully computed with Pearson’s correlation coefficient and visualized in a heatmap, as shown in Figure 4.

It is evident from the results that the potential collinearity among features falls within an acceptable range. This observation supported the assumption that every feature imparts unique and additional information to the classification model, thereby enhancing its overall performance. Considering that the characteristics defined by FHE and PHE are identical in practice (namely, the frequency or position of half energy), we chose to retain only PHE (instead of FHE) because its correlations with most other features are lower in the context of this categorization research. Therefore, only 12 features were ultimately used for further fach classification.

Looking at the classification performance for each voice type, Figure 5 presents the values of the confusion matrices, visualizing the prediction results of the RF model for the test dataset.

From this observation, the classification accuracy for dramatic sound samples was generally high in all voice types. However, the model performed differently for the lyric voice structure. While sopranos and tenors showed the best performance for lyric sound samples, the prediction accuracies for mezzo-sopranos and basses were lower.

Using the established evaluation metrics, the performance of our classification model applied to a range of voice types is given in Table 4, which provides a brief summary of the effectiveness of our model. As shown, our model maintained an accuracy of over 70% across all voice types, with the tenor classification achieving the peak at 82.05%. These results illustrated the consistent predictive ability of our model. The BER demonstrated more disparity, ranging from low rates for sopranos, tenors and baritones to significantly higher rates for mezzo-sopranos and basses. For these two voice types, the elevated BER pointed to a less balanced performance of the model in this particular classification.

According to the principle of the RF classifier, the importance of each feature is depicted in Figure 6. Each bar on the chart corresponds to one of the 12 features considered in our model. The height of a bar indicates the degree of importance a feature holds in the classification process for the respective voice type.

To investigate the role each feature plays in the predictions of the model, we displayed SHAP values for the class ‘dramatic’ in each voice type in Figure 7. This provided further insight into the relative importance of all 12 features for the prediction model. In essence, the SHAP values indicated the extent of influence each feature exerts on the model’s prediction. A positive SHAP value suggests that the presence of a feature pushes the prediction higher, implying a stronger association with the predicted class, while a negative SHAP value means the feature tends to push the prediction lower, suggesting a strong association with the other class.

## 4. Discussion

The developed classification system was able to successfully decide whether a sound sample has a lyric or dramatic character. The results varied according to the voice types. Predictions were excellent for soprano, tenor and baritone voices. Accurate detection rates of about 80% and BER of about 20% can be considered very good, since the discriminatory power of this classification is not very high. This is proven by the fact that there are singers who perform both dramatic and lyric roles, such as the world-famous tenor Luciano Pavarotti [47] (pp. 462–464). Another factor is that there are so-called “Zwischenfächer”, e.g., cavalier baritone [34] (p. 926). This illustrates that the transition between both voice structures can be fluid. On the other hand, close attention must be paid to whether the demands of the respective fach on the vocal folds are manageable for the performers [48,49,50,51]. This applies in particular to the frequently observed development from the lyric to the dramatic fach [33]. In addition, signs of voice changes may be associated with aging [2,52]. That is true, for example, for the parameter VR. On average, dramatic voices have a slower vibrato than lyric ones, but the slowing down of the vibrato is also an age-related phenomenon [27].

For mezzo-sopranos, the classification results were somewhat worse, with an accurate detection rate of about 70% and a BER of about 30%. The spectral timbre parameters SC and FHE confirmed the tendency for lower voice types to have lower values than higher voice types [28]. Traditionally, there is a clear fach distinction in mezzo-sopranos [34] (pp. 931–932), which may well be perceived auditorily in practice. However, our timbre parameters were low for both voice structures; their mean differences were small and statistically not significant. This impaired the fach classification as well, since the ML algorithm could not properly discriminate between lyric and dramatic voice structure for these spectral features. Moreover, the SC and PHE means of lyric mezzo-sopranos were close to those of dramatic sopranos. An overlap of the 95% RR was found for the easily comprehensible and communicable parameter FHE, which can be well translated into the figurative language of vocal pedagogy as well as vocal arts medicine in the context of professional singers’ counseling [53]. Thus, fach changes from lyric mezzo-sopranos to dramatic sopranos like in the famous singer Waltraud Meier [54] (pp. 2157–2160) can be traced with objective parameters. This fach change was also accompanied by a change in voice type. The interaction of the categories voice type and fach is a challenge for objective analysis as well as for auditory perception.

In bass voices, the large difference between BER and accuracy is due to the considerable sample imbalance between the two investigated voice structures. The dark color of all bass voices makes it impossible to classify them with the features used in this study [27]. Therefore, the classification for the bass voice type is not pursued further. Kloiber also does not use the categories lyric/dramatic when describing the fach for bass voices. In contrast, he writes that the division into the categories lyric fach–Zwischenfach–dramatic fach “can be carried out in all voice types” [34] (p. 925). However, the terms “lyric” and “dramatic” are not mentioned in the concrete naming of bass roles, in contrast to all other voice types.

The meticulous methodology of our study facilitated the attainment of reliable and insightful classification outcomes, contributing significantly to our understanding of the delineation between lyric and dramatic voice structure. The RF classifier has demonstrated a generally satisfactory performance in fach classification. One of the key strengths of RF is its robustness to overfitting, thanks to the ensemble learning technique [55]. This model builds multiple decision trees and combines their results, which helps to create a more generalized model that performs well with real-life data [56,57]. Another strength lies in its ability to handle a large number of features without feature scaling or selection. In addition, it provides an estimate of feature importance, which holds instructive significance for vocal arts medicine and singing voice pedagogy. The output of the RF model is also quite interpretable due to its decision tree structure.

However, only basic ensemble learning algorithms were used. Especially in the opera domain, the accuracy of the fach classification can be increased in our classification model. In order to potentially achieve better fach prediction results, more advanced ensemble learning techniques like boosting, stacking, or even integrating deep learning models should be considered in future studies. A further increase in performance is intended and can be achieved by the following means: (1). expanding the amount of tenor, baritone and mezzo-soprano voices in the database to reasonably balance the number of samples regarding voice types and voice structures within a voice type (lyric vs. dramatic), and (2). optimizing the programming of some features. Once these actions have been carried out, the classification results and sound analyses will be supported by an even more solid data basis. This will improve medical advice to opera singers, for example, regarding the following questions:In the phoniatric check-up, how are my current possibilities in solo opera singing assessed with the help of AI-supported evaluation of my objective vocal parameters?Is my voice suitable for a desired change of fach?Are my recent vocal fatigue symptoms due to the selection of my current roles?Are my persistent vocal problems related to my implemented fach change?

To answer these questions, sound samples of new singers are recorded that are comparable in formal structure to the sound samples of our database. Thereafter, the system is trained with all database examples while the new sound recordings are used as test data. The predictions of the system provide the basis for individualized digital voice counseling as an objective aid in the often existential fach classification of opera singers.

## 5. Conclusions

Our contribution describes a novel ML application in the field of medical voice diagnostics that will soon be adopted into our daily practice as specific fach counseling for professional classical opera singers. It provides a clinical decision support system with risk assessment on whether to sing in the lyric or dramatic fach. This approach advances knowledge management and trustworthy artificial intelligence in objective voice diagnostics to prevent vocal damage, professional disability or premature career end in high-performance singers. In the realm of vocal arts medicine and singing voice pedagogy, the enhancement of diagnostic tools and preservation of vocal health remain essential. In future research, we aim to expand the scope of our dataset and delve deeper into advanced ensemble learning algorithms, striving to identify the optimal and most efficient classification methods underpinned by ML techniques.

## Figures and Tables

**Figure 1 diagnostics-13-02870-f001:**
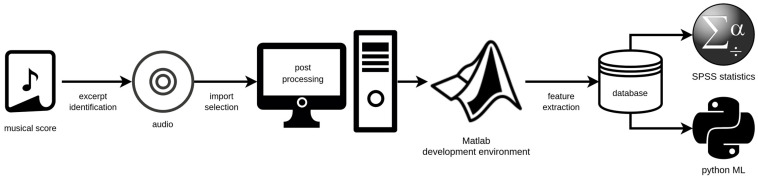
Chart of the overall proposed workflow for fach classification in opera voices.

**Figure 2 diagnostics-13-02870-f002:**
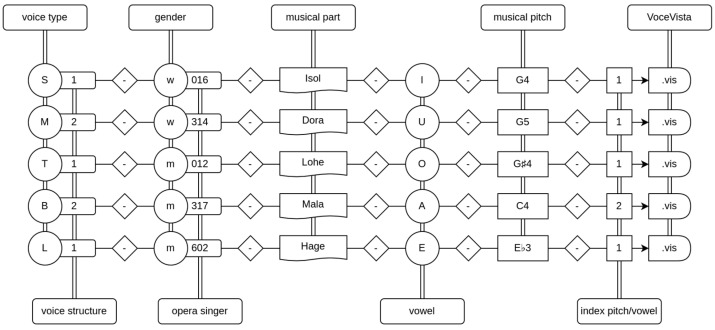
Labeling examples for selected sound samples in every investigated voice type (S—soprano, M—mezzo-soprano, T—tenor, B—baritone, L—bass). Voice structure: 1—dramatic, 2—lyric; gender: m—male, w—female; musical part: Isol—Isolde (from Tristan and Isolde by R. Wagner), Dora—Dorabella (from Così fan tutte by W.A. Mozart), Lohe—Lohengrin (from Lohengrin by R. Wagner), Mala—Maltesta (from Don Pasquale by G. Donizetti), Hage—Hagen (from Twilight of the Gods by R. Wagner); vowel sound symbols according to International Phonetic Alphabet: I—/i:/, U—/u:/, O—/o:/, A—/ɑ:/, E—/e:/; musical pitch according to scientific pitch notation (tuning of concert pitch A4 = 442 Hz): G4–394 Hz, G5–788 Hz, G♯4–417 Hz, C4–263 Hz, E♭3–156 Hz; index pitch/vowel: 1… *n* series number of the analyzed pitch/vowel in case of multiple occurrences.

**Figure 3 diagnostics-13-02870-f003:**
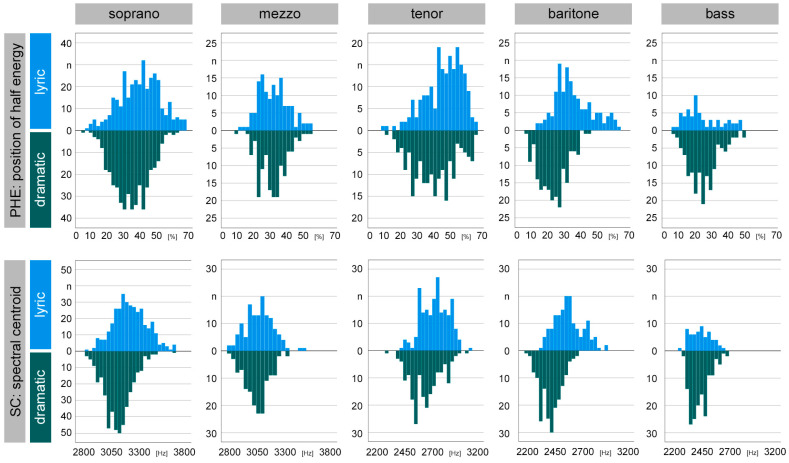
Histograms of timbre parameters PHE (**top**) and SC (**bottom**), sorted by voice type (soprano to bass) as well as voice structure (lyric = blue, dramatic = green). Visually distinct plots when comparing lyric versus dramatic voice structure within a voice type can be easily differentiated (e.g., PHE baritone, SC soprano). Many histograms followed a normal distribution, with the most notable exception being lyric basses, who also suffered from the smallest sample size (*n* = 60).

**Figure 4 diagnostics-13-02870-f004:**
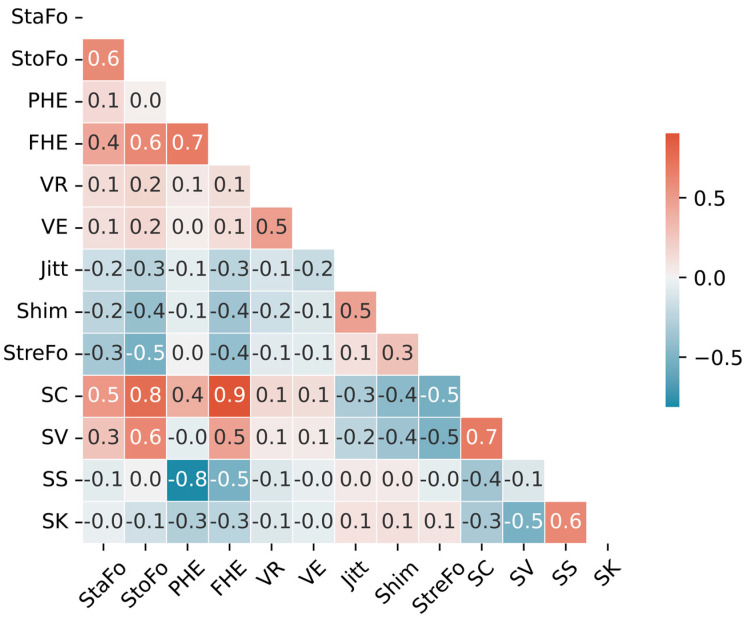
Correlation heatmap of selected acoustic features in voice classification, showing the relationships between candidates for the input features in a RF model.

**Figure 5 diagnostics-13-02870-f005:**
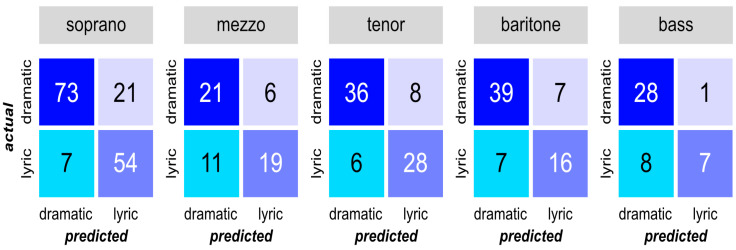
Confusion matrices of all examined voice types for the test dataset.

**Figure 6 diagnostics-13-02870-f006:**
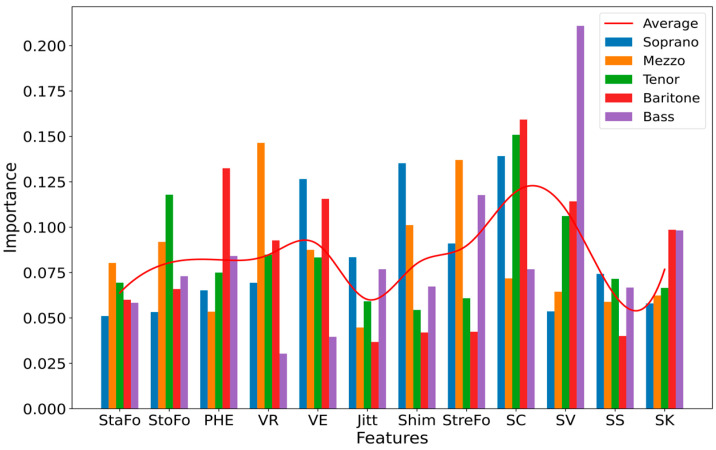
Importance of different features for the classification of voice structure in all investigated voice types, based on impurity.

**Figure 7 diagnostics-13-02870-f007:**
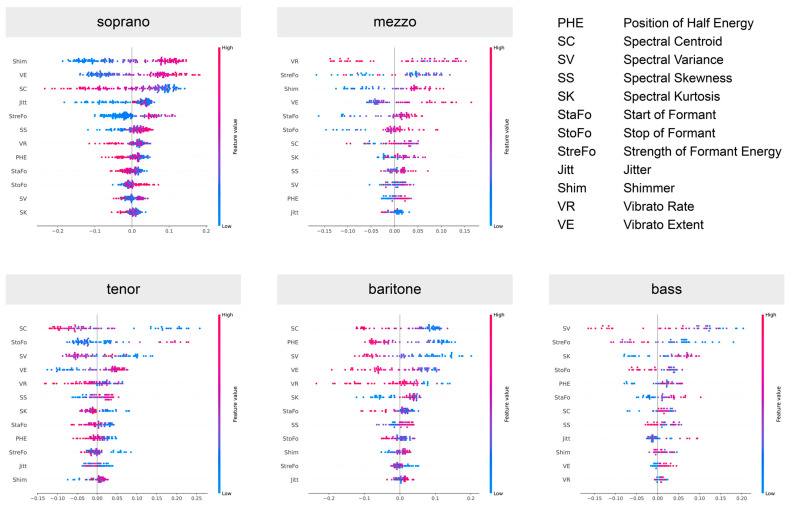
SHAP summary plots of input features for sopranos and mezzo-sopranos (**upper** row), as well as tenors, baritones and basses (**lower** row). Features were sorted in descending order of the impact on the model output. In all plots, only the results for the dramatic voice structure are shown.

**Table 1 diagnostics-13-02870-t001:** Step-by-step instructions for predicting “Fach” labels using the Random Forest classifier.

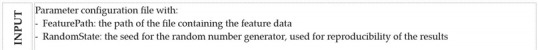
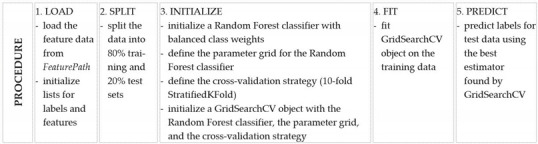
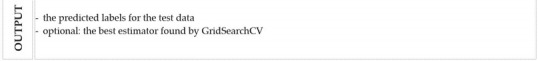

**Table 2 diagnostics-13-02870-t002:** 2 × 2 Confusion matrix visualizing all possible ML model prediction outcomes.

	Predicted Dramatic	Predicted Lyric
Actual Dramatic	True Positive (*TP*)	False Negative (*FN*)
Actual Lyric	False Positive (*FP*)	True Negative (*TN*)

**Table 3 diagnostics-13-02870-t003:** Timbre parameters Frequency of Half Energy (FHE), Position of Half Energy (PHE) and Spectral Centroid (SC) depending on voice structure (lyric vs. dramatic) in all voice types.

Vocal Characteristics of Investigated Opera Singers		Frequency of Half Energy (FHE in Hz)			Position of Half Energy (PHE in %)			Spectral Centroid (SC in Hz)		
Voice Type	Voice Structure (*n* Sound Samples)	Mean (SD)	95% RR	*p*-Value (MWU-Test)	Mean (SD)	95% RR	*p*-Value (MWU-test)	Mean (SD)	95% RR	*p*-Value (MWU-Test)
soprano	all (*n* = 774)	3092 (284)	3072; 3112		35.98 (12.92)	35.07; 36.89		3207 (154)	3196; 3218	
	lyric (*n* = 352)	3169 (307)	3136; 3201	*p* < 0.001	39.46 (13.94)	38.00; 40.92	*p* < 0.001	3271 (159)	3255; 3288	*p* < 0.001
	dramatic (*n* = 422)	3028 (247)	3004; 3052		33.08 (11.23)	32.00; 34.15		3153 (128)	3141; 3166	
mezzo-soprano	all (*n* = 281)	2985 (206)	2960; 3009		31.10 (9.36)	30.00; 32.20		3097 (121)	3083; 3111	
	lyric (*n* = 134)	2990 (217)	2953; 3027	*p* = 0.766	31.36 (9.85)	29.68; 33.04	*p* = 0.767	3110 (132)	3088; 3133	*p* = 0.096
	dramatic (*n* = 147)	2979 (196)	2947; 3011		30.86 (8.92)	29.41; 32.31		3085 (108)	3067; 3103	
tenor	all (*n* = 389)	2705 (221)	2683; 2727		44.05 (13.82)	42.67; 45.43		2740 (143)	2725; 2754	
	lyric (*n* = 199)	2760 (205)	2732; 2789	*p* < 0.001	47.49 (12.77)	45.70; 49.28	*p* < 0.001	2789 (129)	2771; 2807	*p* < 0.001
	dramatic (*n* = 190)	2648 (224)	2616; 2680		40.45 (13.99)	38.45; 42.45		2688 (139)	2668; 2708	
baritone	all (*n* = 343)	2454 (206)	2432; 2476		28.34 (12.86)	26.98; 29.71		2532 (144)	2516; 2547	
	lyric (*n* = 157)	2576 (206)	2543; 2608	*p* < 0.001	35.94 (12.86)	33.91; 37.97	*p* < 0.001	2625 (135)	2604; 2646	*p* < 0.001
	dramatic (*n* = 186)	2351 (140)	2331; 2371		21.93 (8.72)	20.67; 23.20		2453 (97)	2439; 2467	
bass	all (*n* = 217)	2384 (164)	2363; 2406		24.01 (10.24)	22.64; 25.38		2480 (92)	2468; 2493	
	lyric (*n* = 60)	2379 (200)	2327; 2430	*p* = 0.270	23.65 (12.46)	20.43; 26.87	*p* = 0.265	2491 (98)	2466; 2516	*p* = 0.252
	dramatic (*n* = 157)	2387 (149)	2363; 2410		24.15 (9.30)	22.68; 25.61		2476 (89)	2462; 2490	

Legend: MWU = Mann–Whitney U; *n* = number; RR = reference range (lower bound; upper bound); SD = standard deviation.

**Table 4 diagnostics-13-02870-t004:** Classification results in all investigated voice types.

Voice Type	Accuracy	Balanced Error Rate (BER)
Soprano	81.94%	16.91%
Mezzo-soprano	70.18%	29.44%
Tenor	82.05%	17.91%
Baritone	79.71%	22.83%
Bass	79.55%	28.39%

## Data Availability

The dataset used for this study is under a Non-Disclosure Agreement (NDA) and is therefore not available to the public.

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
