# Peer review of "Harnessing Machine Learning in Vocal Arts Medicine: A Random Forest Application for “Fach” Classification in Opera"

_diagnostics, 2023, doi:10.3390/diagnostics13182870_

Round 1
Reviewer 1 Report
The authors have proposed a novel ML application in the field of medical voice diagnostics. This system may be adopted for specific fach counseling. RF algorithm is leveraged to build the ML model. RF is a simple ensemble algorithm. The authors could have gone a bit deeper in making use of powerful ensemble algorithms such as XGBoost, Prophet by Facebook, etc. They have shown that their method has achieved accurate detection rates of about 80% and BER of about 20%. These results can be further improved with advanced ML algorithms.
The authors have touched upon the emerging concept of explainable or interpretable AI. They have leaned on SHAP tool to give a proper explanation about the RF classifier's prediction. Through this, they have articulated the feature importance.
Reviewer 2 Report
This paper proposed singers an improved, objective fach counseling using digital sound analyses and Random Forest algorithm, to advance diagnostic tools and health in vocal arts medicine and singing voice pedagogy. This work is important and the proposed approach is promising. From my point of view, I can see merits and flaws in the paper. Therefore, I suggest a revision of the paper.
My detailed comments are as follows.
1. Author should check the grammar and typo errors and correct them.
2. The innovation of the proposal is not well clarified. What are the open issues? Why do existing solutions fail to address the open challenges? Please better clarify in the introductory section the innovative aspects addressed by your work.
3. A whole figure or flowchart of the overall proposed method is necessary for better understanding.
4. Did perform any pre-processing or post-processing on the audio files? If yes, it is necessary to explain it.
5. Author should explain the reason for using the Mann-Whitney U test. The results of the test should be listed in a table for better understanding.
6. Author should present the results of the confusion matrix.
7. A comparison table with previous literature should be added to understand the strength of this work.
8. Conclusion should also focus on future work.
Minor editing of English language required.
Round 2
Reviewer 1 Report
The authors have meticulously acted on the reviewers' comments and clarifications. The technical contributions of the paper are good and hence the paper is recommended for publication.
The readability of the paper has seen a solid improvement in this version. They have addressed the indicated language issues with all the sincerity and sagacity.